# Identification of NAPRT Inhibitors with Anti-Cancer Properties by In Silico Drug Discovery

**DOI:** 10.3390/ph15070848

**Published:** 2022-07-10

**Authors:** Moustafa S. Ghanem, Irene Caffa, Alberto Del Rio, Jorge Franco, Marco Daniele Parenti, Fiammetta Monacelli, Michele Cea, Amr Khalifa, Aimable Nahimana, Michel A. Duchosal, Silvia Ravera, Nadia Bertola, Santina Bruzzone, Alessio Nencioni, Francesco Piacente

**Affiliations:** 1Department of Internal Medicine and Medical Specialties, University of Genoa, 16132 Genoa, Italy; moustafa.ghanem@edu.unige.it (M.S.G.); irene.caffa@unige.it (I.C.); jorge.franco@innovamol.com (J.F.); fiammetta.monacelli@unige.it (F.M.); michele.cea@unige.it (M.C.); amr.khalifa@edu.unige.it (A.K.); 2Innovamol Consulting Srl, 41126 Modena, Italy; alberto.delrio@innovamol.com; 3National Research Council (CNR), Institute of Organic Synthesis and Photoreactivity (ISOF), 40129 Bologna, Italy; marcodaniele.parenti@isof.cnr.it; 4IRCCS Ospedale Policlinico San Martino, 16132 Genoa, Italy; 5Service and Central Laboratory of Hematology, University Hospital of Lausanne, 1015 Lausanne, Switzerland; aimable.nahimana@chuv.ch (A.N.); michel.duchosal@chuv.ch (M.A.D.); 6Department of Pharmacy, Biochemistry Lab, 16132 Genoa, Italy; silvia.ravera@unige.it (S.R.); nadia.bertola@gmail.com (N.B.); 7Department of Experimental Medicine, University of Genoa, 16132 Genoa, Italy; santina.bruzzone@unige.it (S.B.); francesco.piacente@unige.it (F.P.)

**Keywords:** NAPRT inhibitors, cancer metabolism, NAD, anti-cancer agents, NAMPT, NAD synthesis, in silico drug design

## Abstract

Depriving cancer cells of sufficient NAD levels, mainly through interfering with their NAD-producing capacity, has been conceived as a promising anti-cancer strategy. Numerous inhibitors of the NAD-producing enzyme, nicotinamide phosphoribosyltransferase (NAMPT), have been developed over the past two decades. However, their limited anti-cancer activity in clinical trials raised the possibility that cancer cells may also exploit alternative NAD-producing enzymes. Recent studies show the relevance of nicotinic acid phosphoribosyltransferase (NAPRT), the rate-limiting enzyme of the Preiss–Handler NAD-production pathway for a large group of human cancers. We demonstrated that the NAPRT inhibitor 2-hydroxynicotinic acid (2-HNA) cooperates with the NAMPT inhibitor FK866 in killing NAPRT-proficient cancer cells that were otherwise insensitive to FK866 alone. Despite this emerging relevance of NAPRT as a potential target in cancer therapy, very few NAPRT inhibitors exist. Starting from a high-throughput virtual screening approach, we were able to identify and annotate two additional chemical scaffolds that function as NAPRT inhibitors. These compounds show comparable anti-cancer activity to 2-HNA and improved predicted aqueous solubility, in addition to demonstrating favorable drug-like profiles.

## 1. Introduction

Nicotinamide adenine dinucleotide (NAD) is broadly involved in fundamental biological processes inside the cell. It is particularly unique in its ability to function not only as a cofactor in redox reactions that are intimately involved in energy metabolism but also as a substrate for NAD-consuming enzymes, including poly(ADP-ribose) polymerases (PARPs), sirtuins, CD38, and CD157 [1,2,3,4,5]. Since NAD gets degraded by the catalytic activity of these enzymes, continuous NAD production is required. Most mammalian tissues generate NAD starting from nicotinamide (NAM) through the salvage pathway (also known as the ″amidated″ pathway). A parallel NAD-generating route named the Preiss–Handler (PH) pathway (or ″deamidated″ pathway) utilizes nicotinic acid (NA) as its building block and also operates in many tissues [6,7,8]. In addition, NAD can be synthesized from the amino acid tryptophan through the de novo pathway, which is mainly active in hepatic and renal tissues [9,10]. It is also worth mentioning that nicotinamide riboside (NR) and its reduced form (NRH) were recently recognized as additional NAD precursors that boost NAD production through alternative salvage pathways [11,12,13].

Cancer cells extensively rely on these NAD-biosynthetic routes (summarized in Figure 1) in order to keep adequate NAD levels, which, in turn, are necessary to fuel their reprogrammed metabolism [14] and to compensate for the extensive NAD breakdown caused by vital, NAD-consuming, enzymatic activities, such as PARP-mediated DNA repair [1,2,3,4]. Accordingly, interfering with the NAD biosynthetic machinery has been put forward as an appealing therapeutic approach against cancer. Most studies in this field focused on interrupting the NAM salvage pathway by targeting its rate-limiting enzyme, nicotinamide phosphoribosyltransferase (NAMPT). This reflects the fact that potent and highly active (at least in preclinical models) NAMPT inhibitors, such as FK866 and CHS828, were among the first NAD-lowering agents to be reported [15,16,17] and the observation that NAMPT is commonly overexpressed in a variety of human cancers [17,18]. Regrettably, despite their efficacy in preclinical models, NAMPT inhibitors showed poor efficacy in clinical trials [19,20,21,22], indicating that tumor cells exploit surrogate NAD-producing routes, in particular the PH pathway, to circumvent NAMPT blockade [23,24].

In this context, the relevance of nicotinic acid phosphoribosyltransferase (NAPRT), one of the key enzymes from the PH pathway, as a viable antitumor target has gathered growing attention. NAPRT boosts and regulates NAD biosynthesis under certain conditions and in specific tissues [8,25]. Several studies indicated that the therapeutic activity of NAMPT inhibitors is largely dictated by the NAPRT expression status of the tumor cells [26,27,28]. Susceptibility to NAMPT inhibitors was particularly noted in epithelial-to-mesenchymal transition (EMT)-subtype gastric cancers, in isocitrate dehydrogenase 1 (*IDH1*)-mutant gliomas, and in protein phosphatase Mg^2+^/Mn^2+^-dependent 1D (*PPMD1*)-mutant gliomas as a result of the epigenetic loss of the *NAPRT* gene expression that frequently accompanies these cancer subtypes [29,30,31]. NAPRT was found to be amplified in a large subset of solid human cancers such as ovarian, pancreatic, and breast cancers [23,24]. Accordingly, we showed that several ovarian cancer cell lines became responsive to FK866 upon *NAPRT* knock-down both in vitro and in vivo (in mice ovarian cancer xenografts) [23]. We also demonstrated that NAPRT plays a central role in energy metabolism, DNA repair, and in protein synthesis in cancer cells via its ability to promote NAD production [23]. Based on the above findings, the development of NAPRT inhibitors holds promise for potential application as anti-cancer agents.

Very few NAPRT inhibitors have been reported so far. Early studies on human platelets discovered several compounds that displayed NAPRT inhibitory activity, including 2-hydroxynicotinic acid (2-HNA) and several non-steroidal anti-inflammatory drugs (NSAIDs) such as flufenamic acid, mefenamic acid, and phenylbutazone (Table 1) [32,33,34]. We demonstrated that 2-HNA was indeed able to sensitize NAPRT-expressing ovarian and pancreatic cancer cells to NAMPT inhibitors and recapitulated the effect of *NAPRT* silencing [23]. A series of endogenous NAPRT-inhibiting metabolic intermediates were also identified, with CoA being the most potent metabolite [35]. To our best knowledge, 2-HNA remains the only reported NAPRT inhibitor with proven anti-cancer activity, although its clinical use is limited by poor aqueous solubility. In this work, we performed a high-throughput molecular docking screen with the aim of identifying chemical scaffolds with inhibitory activity on the human NAPRT enzyme. Hence, we were able to identify and characterize as NAPRT inhibitors two compounds that show favorable drug-like properties and potency in the micromolar range.

## 2. Results

### 2.1. Structure-Based Virtual High-Throughput Screening

#### 2.1.1. Analysis of Available 3D Structures of NAPRT

At the time of this work, only one X-ray structure of human NAPRT was available in the public domain (PDB accession code 4YUB), in its ligand-free form [36]. This NAPRT structure was solved at a resolution of 2.9 Å, which usually allows for the unambiguous assignment of the main chain and side chains for the rigid parts of a protein, although a chance of an incorrectly placed side chains still exists. This potential issue, which, at least in principle, could affect the results of subsequent docking studies, was addressed by applying standard protein preparation procedures, such as restrained energy minimization (see methods section). The crystal structure of human NAPRT (Figure 2A) reveals that its monomer folds into 17 α-helices, 24 β-strands, and the connecting loops organized in two domains: a first domain characterized by an irregular α/β barrel and a second open-faced sandwich domain. The structural organization of human NAPRT is highly similar to that of bacterial NAPRT, e.g., the enzyme expressed by T. acidophilum or E. faecalis, for which the X-ray structures are also available (PDB codes 1YTD/1YTE/1YTK and 2FTF, respectively). Despite the low sequence identity (34%), many active site residues are conserved among the different species, suggesting a very similar mode of binding for substrates. The model also reveals the presence of an intimately associated dimer in the asymmetric unit; the two monomers are arranged head to tail with the N-terminal domain in one monomer contacting the α/β barrel in the other monomer (Figure 2A). When structurally compared, the structures of the two monomers showed similarity in terms of the overall protein conformation but slight differences in terms of the shape of the active site; in fact, some residues in the active site, such as Leu170, Arg171, Arg172, His213, and Tyr21, translated into a different conformation between the two protein chains, leading to a significant difference in active site shape and volume (Figure 2B).

#### 2.1.2. Virtual Screening Procedure

The crystal structure of human NAPRT was used as a template for our virtual screenings, which were aimed at identifying compounds with inhibitory activity on NAPRT. This structure was prepared with standard preparation procedures that include the correct assignment of bond orders, adding hydrogen, the optimization of protonation states of residues, and restrained energy minimization. Although some water molecules are present in the NAPRT crystal structure, none of these seem to be involved in stable interactions with active site residues; therefore, all water was removed. To maximize the probability of identifying active molecules, two different high-throughput virtual screening procedures were carried out: 1) a functional dimeric model of human NAPRT was taken as the docking target and 2) the single NAPRT monomer was taken as the docking target. For model 1), in order to tackle the differences in active site conformation (see Figure 2), both active sites identified in the functional dimeric model were used as targets, applying the so-called ensemble docking technique. This strategy allows the docking of a single ligand library against multiple rigid receptor conformations and the combining of the results.

In both models, the HTS Compound Collection from Life Chemicals (https://lifechemicals.com/screening-libraries/hts-compound-collection, accessed on 23 March 2016) consisting of 537,009 drug-like compounds, was docked into a docking grid of 18 Å centered on the active site residues, as shown in Figure 3. Docking results were ranked based on the score, and the first 500 hits were visually inspected to prioritize compounds that reproduced, at least in part, the putative binding mode of the NAPRT substrates. This evaluation led to a final list of 35 purchasable compounds to be tested in vitro as putative NAPRT inhibitors. In addition, from the same Life Chemicals compound collection, a set of 2-hydroxynicotinic acid (2-HNA) analogs (Figure 4) was manually selected, as 2-HNA is known to inhibit NAPRT in the micro/millimolar range of concentration. A brief description of the compounds selected for in vitro characterization can be found in Table 2.

### 2.2. Biological Annotation of the Selected Compounds

#### 2.2.1. In Vitro Compound Screening

To rapidly screen the 50 selected compounds for their ability to inhibit NAPRT, we used their capacity to sensitize the NAPRT-proficient ovarian cancer cell line, OVCAR-5, to FK866 as a reading frame (since this cell line is normally resistant to the NAMPT inhibitor but becomes sensitized to it through either NAPRT silencing or inhibition). By itself, the addition of putative NAPRT inhibitors is postulated to show minimal anti-proliferative activity due to the ability of the cells to use the NAM that is present in the cell culture media to synthesize NAD [23]. OVCAR-5 cells were treated with the putative NAPRT inhibitors at 100 µM concentration with or without 100 nM FK866. As depicted in Figure 5A,B, five compounds out of the 50 that were tested (i.e., compounds **1**, **2**, **8**, **16**, and **19**) led to significant cancer cell growth inhibition when coupled with FK866 while being minimally active when used alone. The remaining compounds were discarded since they were either completely inactive or caused remarkable anti-proliferative activity without FK866 (as observed in Figure 5A with compound **5** and compound **7**). The complete inactivity of some compounds could be ascribed to their inability to bind NAPRT or to poor cell membrane permeability. The intrinsic anti-cancer effect of some of the compounds (i.e., without FK866) was considered to be indicative of non-specific toxicity that would possibly also affect healthy cells.

Afterwards, we aimed at assessing the downstream effects of inhibiting both enzymes in cancer cells, particularly in terms of intracellular NAD concentration. In line with our previous observations with NAPRT silencing, by themselves 2-HNA and the new 5 putative NAPRT inhibitors failed to reduce intracellular NAD levels [23]. However, 2-HNA and the new putative inhibitors did cooperate with the NAMPT inhibitor, FK866, to blunt intracellular NAD concentrations (Figure 5C). We next evaluated the ability of these compounds to sensitize OVCAR-5 cells to lower concentrations of FK866. Four out of the five compounds (i.e., compounds **1**, **2**, **8**, and **19**) were indeed able to sensitize OVCAR-5 cells when incubated (at 200 µM) with increasing concentrations of FK866 (Figure 5D). The degree of sensitization varied among the putative inhibitors, with compound **8** exhibiting the most potent sensitization effect. Notably, the sensitizing activity of compound **8** was even more pronounced than that of the classical NAPRT inhibitor, 2-HNA (Figure 5D). On the other hand, compound **16** was the only compound that completely failed to sensitize the ovarian cancer cells to FK866, and thus it was not further investigated.

To further confirm the observed sensitization effect of our putative inhibitors on the anti-tumor activity of NAMPT inhibitors, we extended our experiments in two additional NAPRT-expressing cancer cell lines (i.e., HCT116 and OVCAR-8). Consistent with our previous observations in OVCAR-5 cells, compound **8** and compound **19** also sensitized these other two cell models to FK866 when they were used at 100 µM concentration (Figure 6A–C). By contrast, compound **1** and compound **2**, when used at the same concentration, failed to sensitize these two cancer cell lines to FK866, with compound **2** even showing unspecific anti-proliferative activity in these models (Figure 6D,E). Since compound **8** (4-hydroxynicotinic acid) and 2-HNA are structural isomers, we decided to evaluate whether the remaining 2-HNA analogs [i.e., 5-hydroxynicotinic acid (5-HNA) and 6-hydroxynicotinic acid (6-HNA)] are also capable of inhibiting NAPRT. We tested this hypothesis in OVCAR-8 cells. Neither 5-HNA nor 6-HNA could recreate the effects of compounds **8** and 2-HNA in terms of cell sensitization to FK866 (Figure 6F). We hypothesize that this reflects the inability of these compounds to bind within the NAPRT enzymatic pocket. Overall, these findings highlight the specificity of compound **8**, since shifting the position of the -OH group from position 4 to position 5 or 6 entirely abolished their ability to sensitize cancer cells to FK866 (and thus, arguably, to inhibit NAPRT). Ultimately, these experiments indicate that the NAPRT-inhibitory activity of these hydroxylated analogs of nicotinic acid strictly relies on -OH substitution at position 2 or 4 of the pyridine ring.

In the PH pathway, NAPRT catalyzes the transfer of a phosphoribosyl group from phosphoribosyl pyrophosphate (PRPP) to its substrate NA, thereby yielding nicotinic acid mononucleotide (NAMN). The latter is converted into nicotinic acid adenine dinucleotide (NAAD) and, finally, amidated into NAD (Figure 1). In order to confirm that the ability of our new inhibitors to sensitize NAPRT-proficient cancer cells to FK866 is on-target, i.e., due to NAPRT obstruction, we supplemented HCT116 and OVCAR-8 cells with NA or NAMN (at 10 µM) while treating them with our putative NAPRT inhibitors, in the presence or absence of FK866. Both NA and NAMN fully rescued these cells from the marked anti-proliferative effect that was achieved by combining FK866 with 2-HNA, compound **8**, or compound **19** (Figure 6G,H). Taken together, these observations are in line with compound **8** and compound **19** being NAPRT inhibitors.

#### 2.2.2. Biochemical Activity on Recombinant Human NAPRT

Given these results in cancer cells, we evaluated the activity of our candidates on the recombinant human NAPRT protein. If the NAPRT enzyme is efficiently inhibited by our compounds, it is postulated to consume less NA compared to what is observed in the absence of NAPRT inhibitors. As expected, the chromatographic analysis revealed higher NA and lower NAMN amounts when compounds **1**, **2**, **8**, and **19** were added to the reaction mixture, in line with their on-target inhibitory activity (data not shown). We performed enzyme kinetic studies to determine the inhibition constant (K_i_) of our putative inhibitors and decipher the fine mechanism underlying their binding to the NAPRT enzyme (Table 3 and Figure 7). Analysis of the kinetic data (V_max_ and K_m_ in the presence of the different inhibitors) suggests that compounds **1**, **2**, and **19** are un-competitive NAPRT inhibitors, with K_i_ of 2281, 89, and 295 µM, respectively. Compound **8** shows similar potency as compound **19** (K_i_ approximately equals 300 µM) and acts as a competitive NAPRT inhibitor, i.e., it competes with NA for the NAPRT catalytic site, as inferred by the fact that V_max_ was not affected and that K_m_ was increased, in the presence of compound **8** (Figure 7, Table 3). In addition, the fact that compound **8** is one of the structurally closest analogs of 2-HNA and NA also lends support to the proposed mechanism of action of this compound. Due to the low potency of compound **1** on the human NAPRT enzyme (K_i_ equals 2.3 mM), in addition to its limited anti-cancer activity in our cancer cell models, we decided to exclude this compound from further experiments. Despite the promising activity of compound **2** on the purified NAPRT protein and in OVCAR-5 cells at 100 µM concentration when coupled to FK866, a 200 µM concentration of the same compound showed remarkable activity in the absence of FK866 in the same cell line (Figure 5D). Similar activity without FK866 was also seen in OVCAR-8 and HCT116 cells at 100 µM (Figure 6D,E). Thus, in view of this intrinsic toxicity, compound **2** was also excluded from further testing. Nonetheless, the core structures of compound **1** and compound **2** could be a starting point for future compound optimization steps. Ultimately, we decided to focus on compound **8** and compound **19** for our subsequent analyses.

### 2.3. In Silico Solubility Prediction and Pharmacokinetic Characterization

In a previous study involving animal experiments, we were unable to dissolve 2-HNA in saline at the desired concentration for intraperitoneal injections, and, thus, we used its sodium salt as an alternative [23]. Poor water solubility is a major hurdle during the drug development process, especially when a drug is meant to be administered orally or parenterally [37]. It was estimated that approximately 40% of the new chemical entities demonstrate modest solubility in water [37]. Given the promising pharmacological results of our drug candidates compound **8** and compound **19**, we addressed their physicochemical and pharmacokinetic parameters. In order to predict their solubility, we made use of the SwissADME website, a publicly available online computational tool that characterizes physicochemical parameters, ADME properties, and the drug-likeliness of a molecule [38]. Compound **8** and compound **19** possess favorable drug-like properties since they don’t violate Lipinski’s rule of five. Based on 2 out of the 3 predictive models employed by the software to calculate water solubility, we found a 1.64- and a 2.25-fold improvement in the predicted molar solubility of compound **8** compared to 2-HNA (reported as Log(S) in Table 4). Likewise, the molar solubility of compound **19** was higher than that of 2-HNA according to all 3 estimating methods (Table 4). Moreover, compound **8** and compound **19** had high predicted GI absorption. Compound **8** had the same bioavailability score as 2-HNA and a higher bioavailability score than compound **19** (Table 4). Neither of the two chemical entities seemed to be able to cross the blood–brain barrier (BBB). Finally, neither compound **8** nor compound **19** was predicted to be a substrate of the efflux transporter P-glycoprotein (Pgp), which is frequently associated with cancer resistance against chemotherapeutics [39]. Collectively, these results indicate promising pharmacokinetic features for compound **8** and compound **19**.

## 3. Discussion

We previously demonstrated that the prototypical NAPRT inhibitor, 2-HNA, synergizes with FK866 in the killing of NAPRT-expressing cancer cells [23]. Herein, we report on the identification of two additional chemical entities that function as NAPRT inhibitors with antineoplastic activity comparable to 2-HNA and with desirable drug-like features.

From a biological standpoint, our studies of cell growth, cellular NAD content, and the enzymatic activity of purified NAPRT unequivocally confirm that compound **8** and compound **19** indeed inhibit this enzyme. Cell-based assays clearly demonstrated that our best NAPRT inhibitor, compound **8** (4-hydroxynicotinic acid), but not 5-hydroxy or 6-hydroxynicotinic acid, exhibited marked anti-cancer activity when combined with FK866 while showing no significant growth inhibition when used alone. These results are consistent with our previous work, showing that per se NAPRT silencing or inhibition do strongly sensitize NAPRT-expressing cancer cells (such as OVCAR-5, OVCAR-8, as well as other cell lines) to NAMPT inhibitors but by themselves have minor anti-proliferative activity [23]. On the other hand, Chowdhry and colleagues showed that the inducible depletion of *NAPRT* caused the regression of OV4 xenografts (PH-amplified ovarian cancer), implying that NAPRT inhibitors might be effective as single agents in similar cancer models that highly depend on the PH pathway to survive [24]. These differences between our studies and the work by Chowdhry and coworkers could be explained by the different cell lines that were utilized.

Our enzyme kinetics analyses indicate that compound **8**, similar to 2-HNA, acts as a competitive NAPRT inhibitor that competes with NA for its enzymatic binding pocket, whereas compound **19** un-competitively inhibits NAPRT. Furthermore, the specificity of our inhibitors was testified by experiments demonstrating that the chemo-sensitizing activity of our inhibitors was abolished upon supplementing cancer cells with sufficient amounts of the substrate (NA) or the downstream product (NAMN) of the NAPRT enzyme. However, similar to the analyses conducted in the case of NAMPT inhibitors [40,41], additional crystallographic studies of the NAPRT enzyme in complex with one or more of our identified inhibitors are warranted to precisely disclose the binding mode of these compounds and describe their interactions within the enzymatic pocket.

Very recently, we demonstrated that gut microbiota caused leukemia cells to display resistance to FK866-induced cell death in vivo when mice were fed with NAM-rich diets through gut-microbiota-derived NA and the consequent activation of the PH pathway in cancer cells (since bacteria use their enzyme nicotinamidase to convert NAM to NA and thereby interconnect the salvage and the PH pathways) [42,43]. Accordingly, coupling FK866 therapy to our NAPRT inhibitors would presumably reverse the protective effect of the intestinal bacteria and restore the anti-tumor effect of NAMPT inhibitors in vivo. Since the in vitro anti-cancer activity observed upon combining NAMPT and NAPRT inhibitors was abrogated when NA was exogenously added in excess to the culture media, it could be argued that the in vivo activity of this combination therapy might be compromised when NA levels rise considerably in the body, as could happen in response to NA- or NAM-rich diets or to NA supplements (e.g., NA is used in gram doses in cases of dyslipidemia due to its lipid-modifying effects) [44]. Future studies should address whether the compounds we identified as NAPRT inhibitors actually show antitumor activity in vivo and whether conditions characterized by high circulating NA levels actually hamper their efficacy. Further improvements in the affinity of these NAPRT inhibitors will increase their therapeutic potential and also reduce the risk of reduced activity in the presence of high NA availability.

## 4. Materials and Methods

### 4.1. Reagents and Cell Lines

OVCAR-5 and OVCAR-8 cell lines were obtained from the NCI-60 panel. The HCT116 cell line was purchased from ATCC (LGC Standards S.r.l., Milan, Italy). The cells were maintained in RPMI-1640 cell culture medium supplemented with 10% heat-inactivated fetal bovine serum (FBS; Gibco, Waltham, MA, USA) and 1% antibiotics [penicillin (50 units/mL)/streptomycin (50 μg/mL) (Life Technologies Italia, Monza, Italy)]. Cells were incubated at 37 °C in a humidified atmosphere of 5% CO_2_ and 95% air. FK866 was bought from the NIMH Chemical Synthesis and Drug Supply Program. All chemical compounds listed in Table 2 were obtained from Life Chemicals. 5-hydroxynicotinic acid and 6-hydroxynicotinic acid were purchased from Thermo Fisher Scientific. Stock solutions of all the putative NAPRT inhibitors were prepared by dissolving the compounds in DMSO at 100 mM. NA, 2-HNA, and NAMN were obtained from Sigma Aldrich S.r.l.

### 4.2. Sulforhodamine B (SRB)Assay

To evaluate the anti-proliferative activity of the putative NAPRT inhibitors in the presence or absence of FK866, the sulforhodamine B colorimetric assay was employed [45]. OVCAR-5, OVCAR-8, or HCT116 cells were seeded in 96-well plates (2 × 10^3^ cells/well) and incubated overnight at 37 °C in a humidified atmosphere of 5% CO_2_ and 95% air to allow cells to adhere. The day after, the old medium was removed from each well and replaced with a fresh culture medium containing the desired compounds at the indicated final concentrations in triplicate, and the plates were subsequently incubated at 37 °C in a humidified atmosphere of 5% CO_2_ and 95% air for 72 h. Afterwards, cold 50% (*w*/*v*) trichloroacetic acid (TCA) was gently added to each well to fix the cells (final concentration, 10% TCA). The plates were incubated at 4 °C for 20 min, then washed four times with tap water and left to air-dry. Thereafter, a SRB solution (0.057% *w*/*v* in 1% acetic acid) was added to stain the fixed cells, and the plates were shaken for 10 min at room temperature. After staining, the SRB solution was removed, and the plates were washed four times with 1% (*v*/*v*) acetic acid and left to air-dry. To solubilize the protein-bound dye, 100 µL of 10 mM trizma base was next added, and the plates were shaken for 10 min at room temperature. Finally, the absorbance was measured at a wavelength of 515 nM by an automated plate reader (Tecan Infinite^®^ 200 PRO instrument).

### 4.3. Intracellular NAD Levels Measurements

To assess whether the antitumor activity of the newly identified putative NAPRT inhibitors, when combined with FK866, was due to their ability to decrease intracellular NAD levels, we performed NAD measurement as follows: 1 × 10^5^ OVCAR-5 cells were plated in each well of a 12-well plate and left to adhere overnight. The day after, cells were treated with combinations of NAPRT inhibitors and FK866 and incubated at 37 °C in a humidified atmosphere of 5% CO_2_ and 95% air for 24 h. The NAPRT inhibitors were used at 100 μM, except for 2-HNA, which was used at 1 mM, and FK866 at 100 nM. After 24 h, cells were lysed with 0.6 M perchloric acid (PCA) at 4 °C and manually detached by a scrapper. The cell lysates were subsequently collected, transferred to new tubes, and diluted in 100 mM Na_2_HPO_4_ at pH 8. To determine the amount of NAD^+^, we utilized a sensitive cyclic assay that takes advantage of the enzymatic activity of alcohol dehydrogenase [46]. Briefly, 100 μL of the diluted samples were pipetted into a white 96-well plate, followed by the addition of 100 μL of the cycling reaction mixture (100 mM Na_2_HPO_4_, 90 U/mL alcohol dehydrogenase, 10 μM flavinmononucleotide, 2% ethanol, 130 mU/mL diaphorase, 2.5 μg/mL resazurin, and 10 mM nicotinamide). Fluorescence increase was measured every 60 s over 30 min using a fluorescence plate reader (544 nm excitation, 590 nm emission). The NAD content was calculated from a standard curve and normalized against the total protein content that was previously quantified for every test sample using the standard Bradford colorimetric assay (Bio-Rad).

### 4.4. Recombinant Human NAPRT Production and Purification

The coding sequence for human NAPRT was cloned in a pET23a vector to insert an N-terminal His-Tag. The recombinant protein was produced in BL-21 (DE3) *E. coli* cells as follows: a starting culture of 5 mL was grown overnight at 37 °C in Luria–Bertani medium supplemented with 100 µg/mL ampicillin. The day after, the culture was diluted at 1:100 in a fresh medium and incubated at 25 °C with the addition of 1 mM NA. When 0.3-0.4 OD_600_ was reached, protein expression was induced with 1 mM IPTG, and bacteria growth was continued overnight at 20 °C. The day after, bacteria were harvested by mild centrifugation (5000 rpm, 10 min) in a Beckman Coulter J6-HC centrifuge and resuspended in 1/50 of the original volume with an equilibration buffer composed of 100 mM K_2_HPO_4_, pH 7.4, 300 mM KCl, and 5 mM imidazole. The cell suspension was sonicated for 10 min at 10 s intervals to disrupt the bacteria cells, and the crude extract was clarified by centrifugation (6000 rpm, 15 min). The recombinant human NAPRT was purified by His-tag affinity chromatography as follows: the supernatant was batch-mixed for 1 h at 4 °C with a HisPur Cobalt resin (Thermo Fisher Scientific, Pittsburg, PA, USA) on a rotating mixer, then the resin was packed into a chromatographic column. The flow-through and the subsequent 10 mM imidazole wash buffer were discarded. The recombinant protein was finally eluted three times with 1 mL of an equilibration buffer containing 150 mM imidazole. The three elutions were merged, and the protein was concentrated with a Protein Concentrator 10K (Pierce-Thermo Fisher Scientific, Pittsburg, PA, USA). The concentrated protein was dialyzed in a SnakeSkin™ dialysis tubing 10K (Thermo Fisher Scientific) overnight at 4 °C against 50 mM Tris/HCl, pH 7.4, 10 mM KCl, and 1 mM DTT to remove the imidazole of the elution buffer. The dialyzed protein was quantified by spectrophotometer absorbance at 280 nm and stored at 4 °C after the addition of 500 μM PRPP to stabilize the protein structure.

### 4.5. Enzymatic Activity Assays and K_i_ Calculation

To determine if the effects of the putative NAPRT inhibitors on cell viability and NAD content were indeed caused by an inhibition of NAPRT enzymatic activity, we set enzymatic reactions with the recombinant human NAPRT, analyzed NAMN formation by HPLC, and the K_i_ was calculated. For each NAPRT inhibitor, reactions with variable NA concentration (between 10 and 640 μM) and variable inhibitor concentration (between 0 and 1000 μM) were performed at 37 °C for a time in which the amount of the product NAMN did not exceed the 10% of the total NA amount. The reactions were blocked by heating samples at 85 °C for 3 min, and the protein was removed by centrifugation. The clarified reactions were analyzed by HPLC with an XTerra MS C18 Column, 125Å, 5 μm, 4.6 mm × 150 mm (Waters) in 100 mM phosphate buffer pH 5 with a gradient of methanol from 0 to 30%. The initial velocities (V_0_) were calculated and inserted in the Michaelis–Menten equation, and subsequently, the K_i_ of each NAPRT inhibitor was calculated with GraphPad Prism 8.

### 4.6. In Silico Screening of the Life Chemicals HTS Compound Collection

In order to identify new drug-like small molecules with inhibitory properties towards human NAPRT, we performed a high-throughput docking virtual screening of the Life Chemicals HTS Compound Collection into the crystal structure of human NAPRT (PDB code: 4YUB). The protein structure was processed with the Protein Preparation Wizard (Schrodinger Maestro v. 2017-4), and 18 Å grids encompassing the catalytic pocket of human NAPRT were generated for the functional human NAPRT dimer and its monomer. Ligands were standardly prepared with Gypsum-DL and docked into the two generated models of human NAPRT with AutoDock Vina [47,48]. The top-500 ranked binding poses from each virtual screening were rigorously evaluated to prioritize compounds that displayed favorable interactions with key catalytic residues and suitable fitting in the NAPRT active site. Selected docking hit compounds were purchased from commercial sources and tested in vitro as putative NAPRT inhibitors.

### 4.7. Statistics

Statistical analyses were carried out with GraphPad Prism software v. 8 (GraphPad Software). All two-group comparisons were performed using an unpaired *t*-test. *p*-values less than 0.05 were considered statistically significant.

## 5. Conclusions

In summary, we took advantage of in silico drug design techniques to identify two small molecules that selectively inhibit NAPRT. The hit rate observed with our virtual screening procedure is essentially consistent with the hit rate obtained in our previous in silico screenings, including work that led us to discover the first selective SIRT6 inhibitors [49]. These studies further underscore the advantage of virtual screening approaches for drug discovery when compared to the traditional high-throughput screening procedures that are time- and resource-intensive and that typically achieve lower hit rates [50]. Our best candidates, compound **8** and compound **19**, were able to restore the sensitivity of NAPRT-expressing cancer cells to NAMPT inhibitors through NAPRT inhibition. Similar to 2-HNA, they showed anti-cancer activity in the micromolar range. Although a substantial improvement in the potency of NAPRT inhibitors has not been achieved yet, the structural backbones of these two inhibitors lend themselves to future optimization efforts. Lastly, computational analysis supported desirable drug-like and pharmacokinetic features of these agents. Altogether, our study lays the background for further studies of these new NAPRT inhibitors, including in vivo testing in mouse tumor models and further drug optimization steps.

## Figures and Tables

**Figure 1 pharmaceuticals-15-00848-f001:**
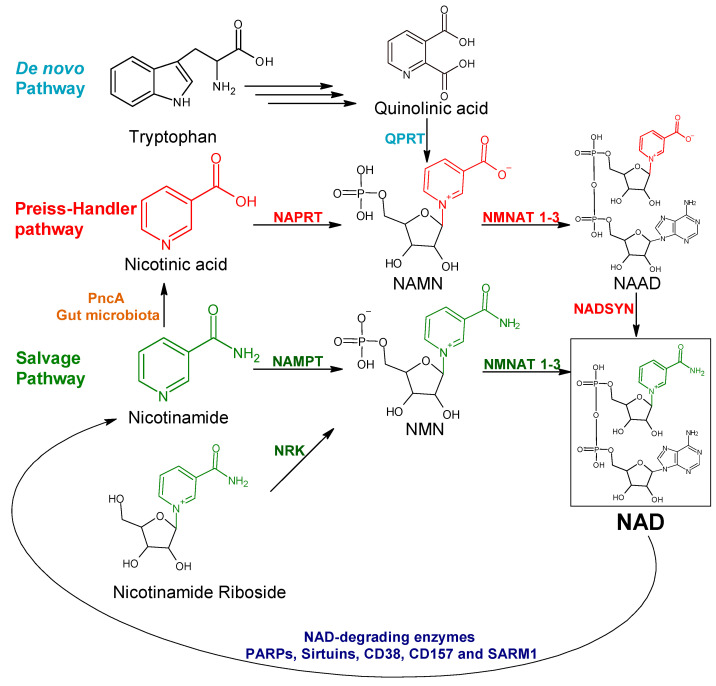
Schematic representation of the NAD–generating pathways in mammalian cells. NAMN, nicotinic acid mononucleotide; NMN, nicotinamide mononucleotide; NAAD, nicotinic acid adenine dinucleotide; NAD, nicotinamide adenine dinucleotide; QPRT, quinolinate phosphoribosyltransferase; NAPRT, nicotinic acid phosphoribosyltransferase; NAMPT, nicotinamide phosphoribosyltransferase; NRK, nicotinamide riboside kinase; NMNAT, nicotinamide mononucleotide adenylyltransferase; NADSYN, nicotinamide adenine dinucleotide synthetase; PncA, nicotinamidase; SARM1, sterile alpha and toll/interleukin receptor [TIR] motif-containing protein 1; PARPs, poly(ADP-ribose) polymerases.

**Figure 2 pharmaceuticals-15-00848-f002:**
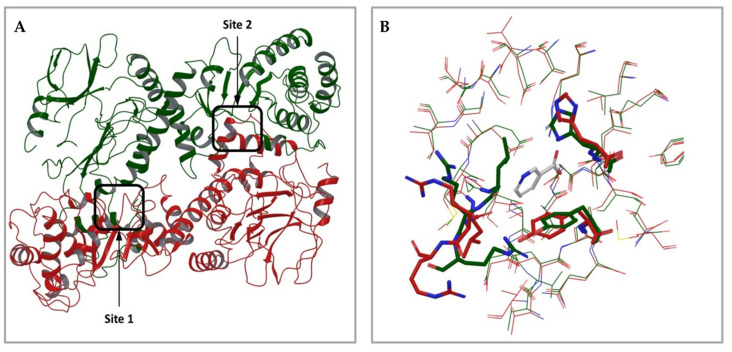
Analysis of the 3D structure of human NAPRT enzyme. (**A**) Overall oligomeric structure of human NAPRT. The enzyme has a dimeric structure and monomers A and B are colored in red and green, respectively. The two active sites are highlighted by black squares. (**B**) Structural superposition of the two active sites in the NAPRT dimer colored red and green, respectively; residues with relevant differences in conformation are drawn in thick tubes.

**Figure 3 pharmaceuticals-15-00848-f003:**
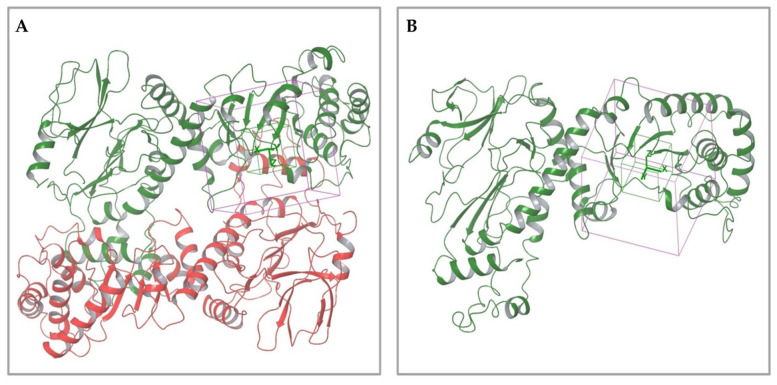
Representation of the NAPRT active site grids employed in docking-based virtual screenings. (**A**) Docking grid of the functional dimeric model of human NAPRT. (**B**) Docking grid of the human NAPRT monomer.

**Figure 4 pharmaceuticals-15-00848-f004:**
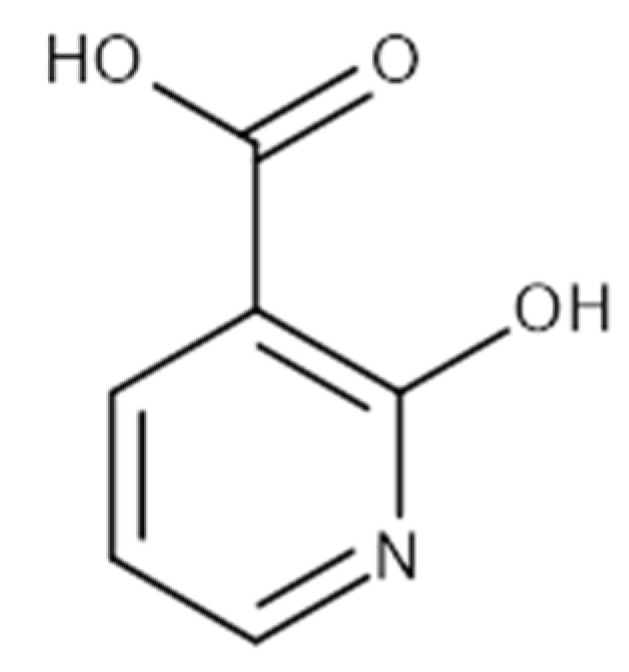
Chemical structure of 2-hydroxynicotinic acid. 2-Hydroxynicotinic acid is the chemical analog of nicotinic acid containing an additional hydroxyl group on carbon 2.

**Figure 5 pharmaceuticals-15-00848-f005:**
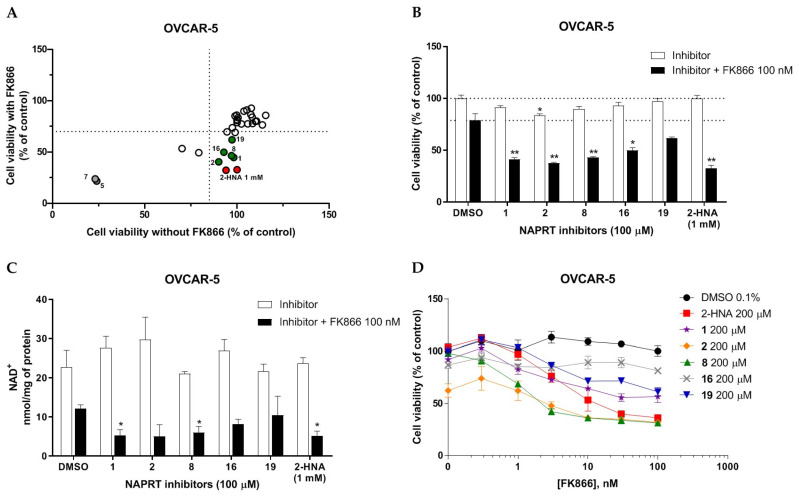
In vitro screening of the putative NAPRT inhibitors. (**A**) Graphical representation of the cell viability results obtained from screening our selected compounds in ovarian cancer cells. OVCAR-5 cells were plated in 96-well plates (2 × 10^3^ cells/well) and left to adhere overnight. The following day, the culture media were replaced with new media containing the respective treatments (i.e., with or without 100 nM FK866 and the putative NAPRT inhibitors, all at 100 μM final concentration, except for 2-HNA, which was used at 1 mM). Each point is the mean of three experimental replicates normalized to the control. The green circles indicate the five most promising putative inhibitors, and the red circles represent 2-HNA as the control NAPRT inhibitor. (**B**) The viability results for the five most-promising NAPRT inhibitors from (**A**) are also represented in a bar graph. *, *p* < 0.05; **, *p* < 0.01 (**C**) OVCAR-5 cells were plated in 12-well plates (1 × 10^5^ cells/well) and allowed to adhere overnight. The following day, the culture media were replaced with new media containing the respective treatments (i.e., with or without 100 nM FK866 and the putative NAPRT inhibitors, all at 100 μM final concentration, except for 2-HNA, which was used at 1 mM). After 24 h, intracellular NAD levels were measured. *, *p* < 0.05 (**D**) OVCAR-5 were plated in 96-well plates (2 × 10^3^ cells/well) and allowed to adhere overnight. The following day, the culture media were replaced with new media that contain the respective treatments (i.e., with or without FK866 at increasing concentrations from 0.3 to 100 nM and the putative NAPRT inhibitors, added at 200 μM final concentration), and the plates were then incubated for 72 h. Afterwards, the cell viability was determined using the sulforhodamine B assay.

**Figure 6 pharmaceuticals-15-00848-f006:**
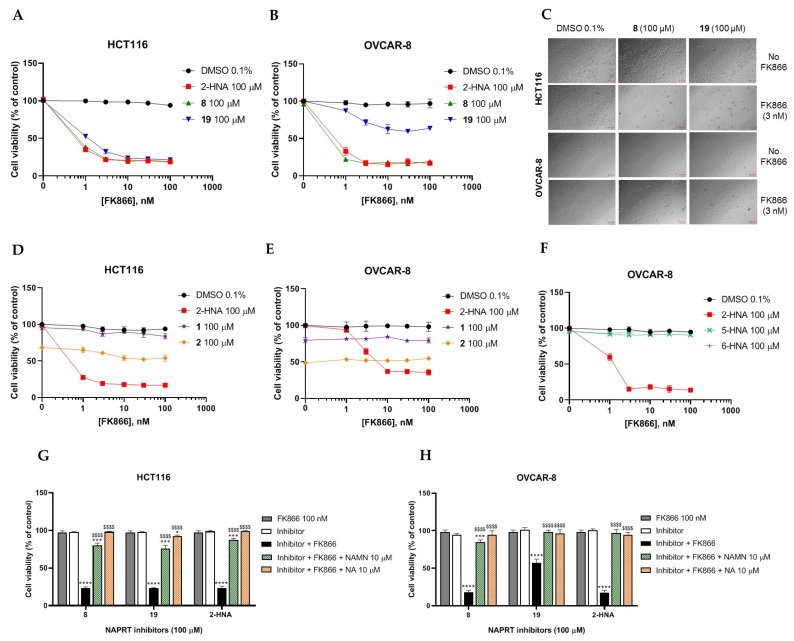
**Compound 8 and compound 19 sensitize ovarian and colon cancer cells to FK866 via NAPRT inhibition.** (**A**–**F**) HCT116 and OVCAR-8 were plated in 96-well plates (2 × 10^3^ cells/well) and allowed to adhere overnight. The following day, culture media were replaced with new media containing the respective treatments (i.e., with or without FK866 at increasing concentrations from 1 to 100 nM and the putative NAPRT inhibitors, added at 100 μM final concentration), and the plates were then incubated for 72 h. Afterwards, the cells were imaged using light microscopy as in (**C**), and cell viability was determined using the sulforhodamine B assay. Data are mean ± SD of three experimental replicates. (**G**,**H**) The same experimental procedure was employed as in (**A**–**F**). Single concentrations of the NAPRT inhibitors (100 µM), 100 nM FK866, 10 µM NA, and 10 µM NAMN were added. Data are mean ± SD of 4 experimental replicates. One representative experiment is shown. *, *p* < 0.05; ***, *p* < 0.001; ****, *p* < 0.0001; $$$$, *p* < 0.0001. The * symbols refer to the statistical significance compared to the treatment with FK866 alone, whereas the $ symbols refer to the statistical significance compared to the combined treatment with FK866 and the NAPRT inhibitors.

**Figure 7 pharmaceuticals-15-00848-f007:**
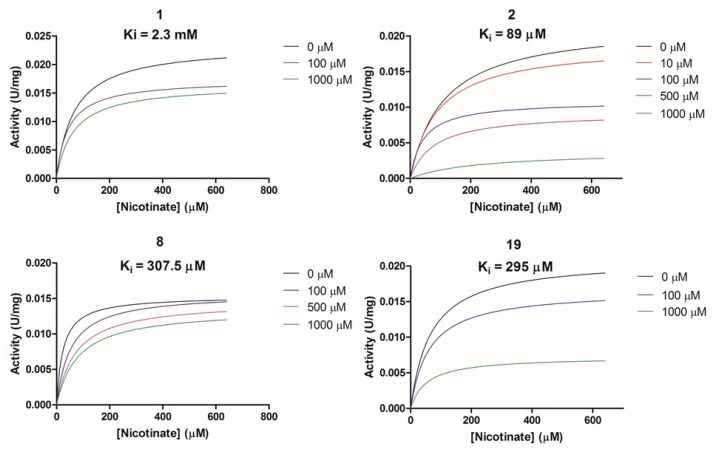
Analysis of NAPRT enzyme activity in the presence or absence of putative NAPRT inhibitors. Graphs represent Michaelis–Menten regression curves of NAPRT reactions performed in the presence of different concentrations of compounds **1**, **2**, **8**, and **19**. The concentration-dependent inhibiting effect on the NAPRT reaction is represented with different colors.

**Table 1 pharmaceuticals-15-00848-t001:** Known inhibitors of human NAPRT enzyme.

Compound	Reported Ki * (µM)	Compound	Reported Ki (µM)
Flufenamic acid	10	6-Chloronicotinic acid	560
Mefenamic acid	50	Isonicotinic acid	750
2-Pyrazinoic acid	75	3-Pyridylsulfonic acid	750
Phenylbutazone	100	Pyridine	780
Indomethacin	150	2-Aminonicotinic acid	820
Salicylic acid	160	Acetanilide	1000
2-Hydroxynicotinic acid	230	Aminopyrine	1000
2-Fluoronicotinic acid	280	Antipyrine	1000
Oxyphenbutazone	300	Picolinic acid	1160
Acetylsalicylic acid	500	3-Pyridylacetic acid	1280
Sulfinpyrazone	500	Benzoic acid	1900

* Ki, inhibition constant.

**Table 2 pharmaceuticals-15-00848-t002:** Selected structurally diverse compounds and 2-HNA analogs for in vitro characterization as putative NAPRT inhibitors.

Compound ID	Structure	Vendor ID	M.W. *	Compound ID	Structure	Vendor ID	M.W.
**1**	** 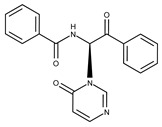 **	F0020-0171	333.3407	**26**	** 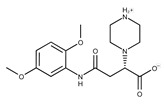 **	F2721-0386	337.3709
**2**	** 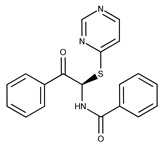 **	F0173-0133	349.4063	**27**	** 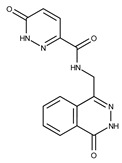 **	F2758-0213	297.2688
**3**	** 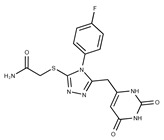 **	F0648-0699	376.3655	**28**	** 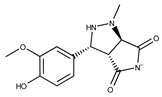 **	F3188-0088	277.2759
**4**	** 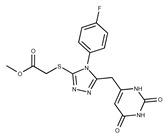 **	F0648-0785	391.3769	**29**	** 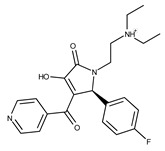 **	F3226-2226	397.4427
**5**	** 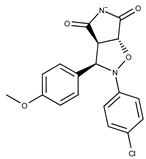 **	F1199-0146	358.7757	**30**	** 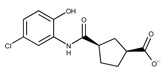 **	F3229-0191	283.7076
**6**	** 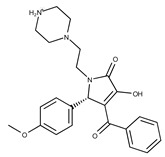 **	F1260-1693	421.4889	**31**	** 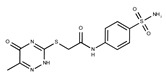 **	F3295-0007	355.3927
**7**	** 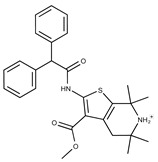 **	F1299-0156	462.6037	**32**	** 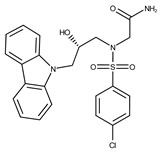 **	F3311-0032	471.9565
**8**	** 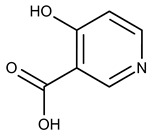 **	F1371-0219	139.1088	**33**	** 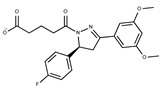 **	F3371-0859	414.4268
**9**	** 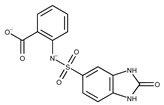 **	F1566-0988	333.3192	**34**	** 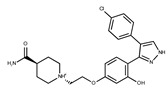 **	F3385-4161	440.9226
**10**	** 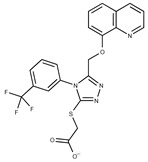 **	F1710-0049	460.429	**35**	** 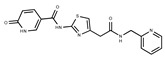 **	F5008-0290	369.3977
**11**	** 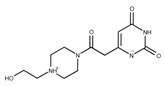 **	F1811-0048	282.2957	**36**	** 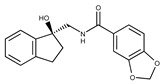 **	F5857-5354	311.3319
**12**	** 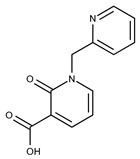 **	F1907-0958	230.2194	**37**	** 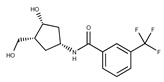 **	F6127-0104	303.2769
**13**	** 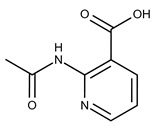 **	F1967s-1157	180.1607	**38**	** 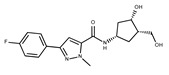 **	F6127-0210	333.3574
**14**	** 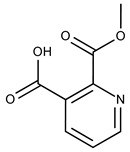 **	F2135-0162	181.1455	**39**	** 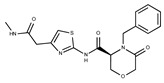 **	F6241-0336	388.4408
**15**	** 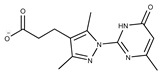 **	F2135-0875	276.2911	**40**	** 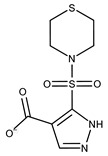 **	F6252-1248	277.3206
**16**	** 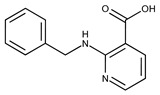 **	F2135-0897	228.2466	**41**	** 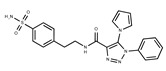 **	F6252-5764	436.4869
**17**	** 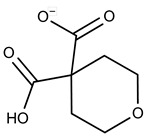 **	F2147-0724	174.1513	**42**	** 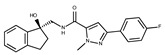 **	F6279-0434	365.4008
**18**	** 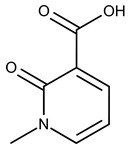 **	F2168-0001	153.1354	**43**	** 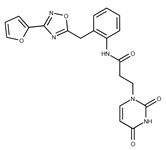 **	F6372-1828	407.3795
**19**	** 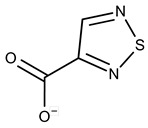 **	F2169-0490	130.1252	**44**	** 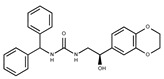 **	F6414-0992	404.4584
**20**	** 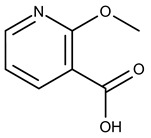 **	F2191-0003	153.1354	**45**	** 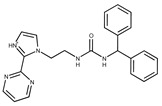 **	F6439-7266	398.4604
**21**	** 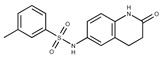 **	F2278-0232	316.3748	**46**	** 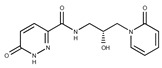 **	F6465-0031	290.2747
**22**	** 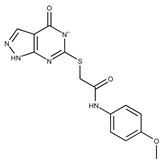 **	F2503-0045	331.3497	**47**	** 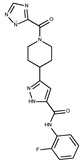 **	F6497-7775	383.3796
**23**	** 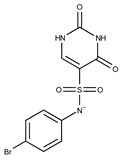 **	F2526-0046	346.1572	**48**	** 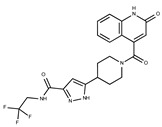 **	F6497-8054	447.4104
**24**	** 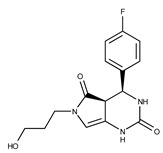 **	F2711-0182	305.3042	**49**	** 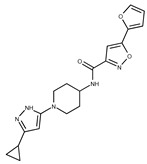 **	F6523-1712	367.4017
**25**	** 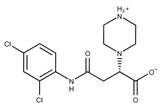 **	F2721-0331	346.2091	**50**	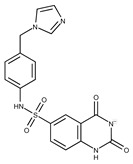	F9994-0201	397.4078

* M.W: Molecular weight.

**Table 3 pharmaceuticals-15-00848-t003:** Proposed mechanism of action for putative NAPRT inhibitors.

Compound ID	Vendor ID	Ki (µM)	V_max_/K_m_	Proposed Mechanism
**1**	F0020-0171	2281	V_max_ ↓/K_m_ ↓	Un-competetive
**2**	F0173-0133	88.99	V_max_ ↓/K_m_ ↓	Un-competetive
**8**	F1371-0219	307.5	V_max_ =/K_m_ ↑	Competitive
**19**	F2169-0490	295.1	V_max_ ↓/K_m_ ↓	Un-competitive

**Table 4 pharmaceuticals-15-00848-t004:** Predicted water solubility and additional pharmacokinetic properties of the two most promising putative NAPRT inhibitors.

Compound ID	Log S (ESOL)	Log *S* (Ali)	Log *S* (SILICOS-IT)	GI Absorption	Pgp Substrate	BBB Permeant	Bioavailability Score
2-HNA	−1.65	−1.97	−0.8	High	No	No	0.85
**8**	−1.44	−1.62	−0.8	High	No	No	0.85
**19**	−1.18	−1.77	−0.26	High	No	No	0.56

## Data Availability

Data is contained within the article.

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
