# Peer review of "Identification of NAPRT Inhibitors with Anti-Cancer Properties by In Silico Drug Discovery"

_pharmaceuticals, 2022, doi:10.3390/ph15070848_

Round 1

Reviewer 1 Report

The manuscript discusses the finding of a molecule that inhibits the activity of NAPRT, which is involved in the cellular activities of cancer cells. The presented works may lead to the generation of new anticancer agents.

However, there are a few issues that the author should put into consideration:

1. The manuscript is lengthy and exhaustive to read; the introduction is more like a review rather than a description of the work. The motivation also became vague due to this lengthy writing. Please screen to include the only necessary information to the work, for example, the paragraph discussing CADD could be removed because it is discussing the method, not about the work (except if the procedure is altered or deviates from the standard strategy).

2. On the structure of the protein model, please add information about its quality. For example, its resolution to judge whether it is rigid or flexible enough and thereby differences in the active shape and volume are indeed significant. If different, what is the implication? Is the enzyme has broad specificity, has a ping-pong reaction mechanism, is able to facilitate 2 different substrates or execute 2 reactions at the same time, or something else? How accessible is the binding pocket to the solvent? If the docking were done to the other active site, would it result in the same target compound? Under what condition the protein was crystallized, was it near to its physiological working conditions or not (e.g. acidic, high salt or organic additive, etc)? The water molecules are indeed usually removed prior to docking and subsequent computational calculation, but their positions are useful to indicate whether the position occupied by the molecule is a possible position for an atom (of the docked molecule) to sit.

3. The term "identification of novel" is dubious. First, the molecules are not designed and (then) synthesized based on the docking but from the HTS list (thus is not novel, the finding of their activity is). Second, the scaffold is known (2-HNA or 5-HNA); compound 8 is 2-HNA isomers and its hydroxyl groups are decorative/variation to the 2-HNA core molecule (the sentences on page 13 lines 266-272 are the main outcome of the in silico work).

4. Some parts of the discussion are repetitive to the introduction or analysis of the result. 

Author Response

  • The manuscript is lengthy and exhaustive to read; the introduction is more like a review rather than a description of the work. The motivation also became vague due to this lengthy writing. Please screen to include the only necessary information to the work, for example, the paragraph discussing CADD could be removed because it is discussing the method, not about the work (except if the procedure is altered or deviates from the standard strategy).

We would like to thank the reviewer for pointing out this issue. We addressed this point by:

  • Removing the paragraph that introduces CADD as suggested by the reviewer.
  • Shortening the introduction so that it contains the most relevant information to the work. After applying the above corrections, the introduction now consists of 4 paragraphs: The first two paragraphs provide a comprehensive overview about the topic while the last two paragraphs focus on our target enzyme NAPRT. This would also help clarifying the motivation of the work to the reader.
  • In addition, the motivation of this work was described in the last 2 sentences of the introduction “In this work, we performed a high-throughput molecular docking screen with the aim of identifying chemical scaffolds with inhibitory activity on human NAPRT enzyme. Hence, we were able to identify and characterize as NAPRT inhibitors two compounds that show favorable drug-like properties and potency in the micromolar range.”.

  • On the structure of the protein model, please add information about its quality. For example, its resolution to judge whether it is rigid or flexible enough and thereby differences in the active shape and volume are indeed significant. If different, what is the implication?

We added a statement in the text to comment on protein structure quality and flexibility.

 “ This  NAPRT structure was solved at a resolution of 2.9 Å, which usually allow for unambiguous assignment of the main chain and side chains for the rigid parts of a protein, although chances of incorrectly placed side chains still exist. This potential issue, which, at least in principle, could affect the results of subsequent docking studies, we addressed by applying standard protein preparation procedure, such as restrained energy minimization (see methods section) “

Is the enzyme has broad specificity, has a ping-pong reaction mechanism, is able to facilitate 2 different substrates or execute 2 reactions at the same time, or something else?

To our knowledge, there is no evidence in literature of reaction mechanism for NAPRT such as ping-pong mechanism as well as for the formation of reaction intermediate complexes with different substrates. Therefore, we believe that no additional comment should be provided in the text.

How accessible is the binding pocket to the solvent? If the docking were done to the other active site, would it result in the same target compound?

The docking was conducted in both putative active sites and results were combined (ensemble docking). A brief statement was added to the text to clarify this procedure.

“For model 1), in order to tackle the differences in active site conformation (see Figure 2), both active sites identified in the functional dimeric model were used as targets, applying the so-called ensemble docking technique. This strategy allows to dock a single ligand library against multiple rigid receptor conformations and combine the results. “

Under what condition the protein was crystallized, was it near to its physiological working conditions or not (e.g. acidic, high salt or organic additive, etc)?

The crystallization procedure was originally described by Marletta et al. (FEBS Open Bio. 2015;5:419-28). In this article, the authors specify that the “crystals of hNaPRTase were obtained by means of the vapor diffusion technique in sitting drops. 4 μL of a protein solution at a concentration of 15 mg/mL, preincubated with 1 mM Acetyl-CoA and 5 mM DTT, were mixed with an equal volume of a reservoir solution containing 0.1 M Sodium cacodylate pH 6.5 and 1.7 M Sodium acetate and equilibrated against 1.2 mL of the reservoir solution, at 4 °C. Crystals grew to maximum dimension in about 5 days.” As reported in our article, at the time of the study this was the only one human NAPRT X-ray structure available in the public domain. Obviously, an X-ray structure represents in some way a limited representation of physiological condition but it is anyway a good starting point to analyze protein structure and design new inhibitors. On the other hand, the fact that, by using this NAPRT crystal that was described by Marletta and colleagues, we were able to actually identify compounds with inhibitory activity on NAPRT (both towards the purified protein and towards the protein naturally expressed in cells) indicates that the crystal itself does closely represent the actual NAPRT structure.

The water molecules are indeed usually removed prior to docking and subsequent computational calculation, but their positions are useful to indicate whether the position occupied by the molecule is a possible position for an atom (of the docked molecule) to sit.

As a general rule, water in the active site of an enzyme could be removed prior to docking simulation to allow the binding of larger ligands; at the same time, free energy changes resulting from displacing water molecules in the active site should be also considered, but the calculation of these parameters require computationally expensive methods. Since no prior knowledge of the binding mode of known ligands or of conserved water molecules in the active site (e.g. from other X-ray structures) is available, we choose to explore all possible variations in the binding pose of putative inhibitors by removing all the water molecules, after an analysis of its interaction network in the active site. The position of removed waters was then compared to binding poses identified by docking to guide the selection of the most interesting molecules.

  • The term "identification of novel" is dubious. First, the molecules are not designed and (then) synthesized based on the docking but from the HTS list (thus is not novel, the finding of their activity is). Second, the scaffold is known (2-HNA or 5-HNA); compound 8 is 2-HNA isomers and its hydroxyl groups are decorative/variation to the 2-HNA core molecule (the sentences on page 13 lines 266-272 are the main outcome of the in silico work).

This point was addressed by: modifiying the title of the article to “Identification of NAPRT Inhibitors with Anti-cancer Properties by In silico Drug Discovery”. In addition, this point was also considered in the text, by stressing that the  novelty of the identified NAPRT inhibitors is related to their activity.

  • Some parts of the discussion are repetitive to the introduction or analysis of the result. 

To avoid repetitions in the discussion, the latter was modified as follows:

  • Some parts of the first paragraph of the discussion were either completely deleted or moved to the introduction.
  • The fourth paragraph of the discussion was also deleted since similar content was described in the results section 2.3. The main message of this paragraph was instead highlighted by one sentence in the conclusion: “Lastly, computational analyses supported desirable drug-like and pharmacokinetic features of these agents.”
  • Section “5. Conclusions” was added to the manuscript and the last paragraph of the discussion that summarizes the work was moved from the discussion to this newly-added conclusions section.

Reviewer 2 Report

Dear authors the work presented for review is very interesting and solidly prepared. However, after reading it, some questions come to my mind:

1. The SRB test determines the effect on cell proliferation/growth. Describing the test results the authors mislead the reader. Please read: https://dtp.cancer.gov/discovery_development/nci-60/methodology.htm

In the cited literature item, a different test was used to assess viability.

2. There is no statistics section in the publication.

3.  Information on culture conditions and test performance should also be included. 

Author Response

  • The SRB test determines the effect on cell proliferation/growth. Describing the test results the authors mislead the reader. Please read: https://dtp.cancer.gov/discovery_development/nci-60/methodology.htm In the cited literature item, a different test was used to assess viability.

We would like to thank the reviewer for pointing out this issue and we apologize for this miscommunication regarding the cited literature item. To avoid misleading the reader:

  • We changed the article that is cited at this point and we now refer the reader to the original article “PMID: 17406391 Sulforhodamine B colorimetric assay for cytotoxicity screening”. We typically use the SRB assay to determine the anti-proliferative effects of NAD-lowering agents since the MTT assay tends to overestimate the activity of these compounds on cell viability (Cea et al. Blood. 2009;113(23):6035-7).
  • In addition, the experimental procedure of the SRB assay was thoroughly described in the methods section 4.

  • There is no statistics section in the publication.

Thanks again for noticing this problem. A statistics section was added in the methods section 4.7

  • Information on culture conditions and test performance should also be included. 

The information on culture conditions and on our test performance were included in the methods sections 4.1, 4.2 and 4.3

Round 2

Reviewer 2 Report

Dear Authors, congratulations on the work done on the preparation of the manuscript and thank you for addressing all comments and for making appropriate changes to the manuscript. All comments have been taken into account and changes have been made therefore, I recommend the manuscript for publication in its current form.